# Research on a Non-Contact Multi-Electrode Voltage Sensor and Signal Processing Algorithm

**DOI:** 10.3390/s22218573

**Published:** 2022-11-07

**Authors:** Wenbin Zhang, Yonglong Yang, Jingjing Zhao, Rujin Huang, Kang Cheng, Mingxing He

**Affiliations:** 1College of Mechanical and Electrical Engineering, Kunming University of Science and Technology, Kunming 650504, China; 2College of Science, Kunming University of Science and Technology, Kunming 650504, China

**Keywords:** non-contact voltage measurement, voltage sensor, electrode array, PCB process, Kalman filtering

## Abstract

Traditional contact voltage measurement requires a direct electrical connection to the system, which is not easy to install and maintain. The voltage measurement based on the electric field coupling plate capacitance structure does not need to be in contact with the measured object or the ground, which can avoid the above problems. However, most of the existing flat-plate structure voltage measurement sensors are not only expensive to manufacture, but also bulky, and when the relative position between the wire under test and the sensor changes, it will bring great measurement errors, making it difficult to meet actual needs. Aiming to address the above problems, this paper proposes a multi-electrode array structure non-contact voltage sensor and signal processing algorithm. The sensor is manufactured by the PCB process, which effectively reduces the manufacturing cost and process difficulty. The experimental and simulation results show that, when the relative position of the wire and the sensor is offset by 10 mm in the 45° direction, the relative error of the traditional single-electrode voltage sensor is 17.62%, while the relative error of the multi-electrode voltage sensor designed in this paper is only 0.38%. In addition, the ratio error of the sensor under the condition of power frequency of 50 Hz is less than ±1% and the phase difference is less than 4°. The experimental results show that the sensor has good accuracy and linearity.

## 1. Introduction

Transmission lines are one of the most critical forms of transmission equipment in the power system. The accurate measurement of transmission line voltage is of great significance to the stable operation and power security of the power system. At present, the mainstream voltage measurement methods mainly include traditional contact voltage measurement and new non-contact voltage measurement [1].

At present, the traditional electric energy information acquisition mainly adopts the contact voltage measurement technology, which requires the metal part of the cable to be in contact with the measurement probe [2,3]. The probe is connected to the secondary equipment for voltage and current measurement. Contact voltage measurement requires that the copper core and the measuring probe are in direct contact, which not only damages the cable insulation layer, but also must be installed without a power supply, affecting the power supply quality and making installation difficult. In the process of contact measurement, the insulation layer of cable is damaged and the operation safety of the power cable and equipment faces severe challenges.

The non-contact voltage measurement (NCVM) technique can measure the electric potential without any electrical contact [4], which is a smart and safe method of voltage measurement [2]. Non-contact voltage measurement technology is of great significance in the internet of things, power electronics, and high voltage engineering [5]. At present, there are two main non-contact voltage sensing methods: photoelectric sensing and electric field coupling sensing [5,6,7,8,9,10,11,12]. Photoelectric sensing technology is based on the optical principle of voltage measurement [13,14,15,16]. This method has good measurement accuracy, but it is difficult to achieve large-scale promotion of engineering applications because of the complex structure of optical devices, high cost, complex daily maintenance and repair, and the fact that it is easily affected by ambient light. Non-contact voltage measurement based on electric field coupling is more convenient for industrial application [12,17,18,19,20,21,22,23]. Its basic principle is that, by applying a certain voltage or electric field, electromotive force will be generated on the induction electrode, and a resistance or capacitance connection can be formed between the electrodes. Based on this principle, different studies have been carried out on non-contact voltage sensors. The authors of [24] designed and implemented a voltage monitoring technology for overhead transmission lines based on contactless capacitive coupling and supplemented by magnetic field sensing. In the work of [25], a voltage acquisition system with a open close coaxial structure is introduced, which allows the measurement of power line voltage waveform without electrical contact. However, the structure is large in size, complicated in process, inconvenient for installation and rotation, and high in cost. The work of [26] analyzes the capacitive non-contact AC voltage measurement technology and the feasibility of measuring arbitrary waveform signals. Several errors related to this technology are analyzed to show the influence of different design parameters on the final accuracy. In order to optimize the design, different sensor structures are studied. In the work of [27], a three-electrode sensor is designed and the electrode is placed in a ring to improve the electric field distribution, insulation performance, and sensor sensitivity [28]. In addition, the exact spatial location of overhead transmission lines is usually unknown and dynamic in practice. The authors of [28] designed a non-contact household voltage measurement device based on the capacitive coupling principle, but this device has the problem of inconvenient installation. Shenil P. S. of Madras Institute of Technology in India and others proposed an instrument scheme suitable for non-contact measurement of AC voltage of insulated conductors. The AC voltage of the insulated conductor can be measured using a suitable probe, which forms a capacitance network between the conductor and the ground, and indirectly calculates the wire voltage by online measuring the voltage across the capacitor [29]. In the laboratory, the prototype of the measurement system was tested at different frequencies and voltages, the data were processed by fast Fourier transform (FFT), and the measurement error was less than 0.75%.

According to Helmholtz theory, the induced electric field of the power line is spatially distributed and this distribution may affect the integration of the electric field, thus generating corresponding probe charges. The sensing accuracy largely depends on the relative position of the power line and the probe.

In a multi-sensor voltage measurement system, the measurement results mainly depend on the processing of the measurement signals of all sensors. The requirement for the signal processing method is to calculate the voltage value to be measured from the measurement result containing a certain measurement error. For the measurement of AC voltage, according to the principle of electric field induction, it can be known that the average value of the sensor output in the sensor ring around the wire is proportional to the wire voltage. Therefore, the voltage value to be measured can be obtained by averaging the output signals of the voltage sensor loop. In addition, according to the characteristics of the spatial electric field distribution around the wire, the output signal of the voltage sensor ring can be spatially Fourier transformed, and the nonlinear equation can be solved according to the analysis result, so as to obtain the value of the voltage to be measured. Kalman filter is a method to obtain the best filter value through signal estimation and correction based on the known system model and noise characteristics [30].

In view of the above analysis, this paper proposes a non-contact voltage sensor structure based on a multi-electrode array and designs a new electrode array structure composed of six uniformly distributed electrodes. The direct contact of the circuit solves the intrusive and safety problems of the touch sensor to a large extent. At the same time, it effectively solves the problem that the accuracy of single sensor non-contact sensor measurement is easily affected by the wire position. The sensor is manufactured using a PCB process, ensuring that the sensor is cheap, accurate, lightweight, and compact. At the same time, a signal processing algorithm for multi-sensor AC voltage measurement based on Kalman filter is also proposed in this paper. According to the electric field distribution around the wire, the state equation of the system is established and the recursive method of the Kalman steady-state filter is given. The Kalman filter algorithm is analyzed from the perspective of algorithm complexity, measurement accuracy, and flexibility, as well as algorithm application conditions and requirements.

The multi-electrode array non-contact voltage sensor designed in this paper has the advantages of convenient use, isolation from the measurement circuit, low insulation design requirements, and convenient installation. It can be used for voltage measurement in low-voltage distribution networks and user terminals. The signal processing algorithm of multi-sensor AC voltage measurement based on the Kalman filter is beneficial to reduce the influence of interference voltage on the measurement and it is not sensitive to the influence of random errors caused by the sensor itself, maintaining high measurement accuracy.

This paper first introduces the research status of non-contact voltage measurement technology and, according to the problem that the measurement accuracy of the existing sensor is easily affected by the relative position of the wire and the sensor, a non-contact voltage sensor structure and data based on a multi-electrode array is proposed. Processing algorithm; the second part mainly introduces the principle of voltage measurement and the parameter design of the sensor; the third part mainly introduces the experimental platform, data processing algorithm, and test results of the sensor test; then, the application scenarios and shortcomings of the sensor are discussed.

## 2. Design of the Voltage Sensor

### 2.1. Principle of the Voltage Sensor

For a long straight wire with an alternating current, an electric field that changes with the internal current of the wire will be generated around the electrified wire and a changing potential will be generated on the conductor around the wire. In addition to the size, shape, position, and other factors of the conductor, the potential on the conductor has a direct relationship with the potential of the electrified wire that generates the electric field. Therefore, using this principle, while fixing other influencing factors, the internal potential of the electrified wire can be measured in a non-contact manner. The traditional voltage contact measurement method is generally to connect the measurement loop into the circuit to be measured and directly collect signals. Different from the traditional method, the principle of the non-contact voltage sensor is to select a metal plate to be coupled with the wire to be measured, form a coupling capacitance at the electrode plate, and collect the induced voltage signal on the coupling electrode plate.

The measurement principle of the non-contact transmission line voltage sensor proposed in this paper is shown in Figure 1. The sensor uses the stray capacitance $C_{l1}$ between the overhead transmission line and the sensor induction plate as the high-voltage arm capacitance. Figure 1 shows the physical process and equivalent circuit of the induced voltage $V_{IN}$ between the internal induced electrode group and the transmission line voltage. It is assumed that the resistor R and the capacitor C are connected between $V_{IN}$ and $V_{R}$, which represents the stray capacitance between the power line and the internal probe group, while $C_{l2}$ represents the stray capacitance between all conductors connected to $V_{R}$ and the ground. The current I flowing through the measuring device is shown in Figure 1.

Therefore, the current I in the circuit can be expressed as follows:(1)$$I=\frac{V_{S}}{\frac{1}{j\omega C_{l1}}+\frac{1}{j\omega C_{l2}}+\frac{R}{1\;+\;j\omega CR}}$$

If the expected stray capacitances $C_{l1}$ and $C_{l2}$ are at a maximum, and if the value of capacitance C is much larger than 10 nf, then the impedance of RC is much smaller than the impedance of stray capacitance, so $\frac{R}{1\;+\;j\omega CR}$ can be ignored, then
(2)$$I=\frac{jV_{S}}{\frac{1}{wC_{l1}}+\frac{1}{wC_{l2}}}$$

Therefore, the voltage is as follows:(3)$$\left(V_{IN}-V_{R}\right)=I\frac{R}{1+j\omega CR}=\frac{j\omega V_{S}R}{\left(\frac{1}{\omega C_{l1}}+\frac{1}{\omega C_{l2}}\right)\left(1+j\omega CR\right)}$$

Voltage $V_{IN}-V_{R}$ is an attenuated version of $V_{S}$ without distortion or phase shift. The attenuation factor γ is as follows:(4)$$\gamma =\frac{C_{l1}C_{l2}}{C\left(C_{l1}+C_{l2}\right)}$$

Its value depends on stray capacitance.

Only when the stray capacitances $C_{l1}$ and $C_{l2}$ are known or can be evaluated, can the amplitude of $V_{S}$ be accurately predicted from $V_{IN}-V_{R}$. However, for power factor and power quality measurement, it is not necessary to know the absolute value of the voltage, but it is important to know the phase and amplitude of the voltage.

### 2.2. Sensor Parameter Design

In order to obtain reasonable structural parameters, it is necessary to verify the influence of wire position change on the measurement results when the number of electrode plates is different. By establishing simulation models to change the wire position under different number of electrode plates, the simulation conditions are set with the number of electrode plates of 1, 2, 4, and 6, respectively, and the wire is moved by 2 mm, 4 mm, 6 mm, 8 mm, and 10 mm in the direction of x, y, and 45° with the positive half axis of x. Considering that the actual measured voltage values of the sensor are 220 V and 380 V, the reference voltage of the wire is uniformly set to 220 V in this simulation. In order to facilitate the installation and use of the sensor, the overall size of the sensor is kept as small as possible during the size design. Therefore, the initial distance between the electrode plate and the wire is 30 mm, which is roughly the same as the actual size.

When only one plate is simulated, the initial distance between the plate and the wire is 40 mm and the floating potential sensed by the plate is 85.00 V. Then, the wire is shifted by 10 mm in the positive direction of the y-axis. At this time, the starting distance between the plate and the wire is 50 mm and the floating potential sensed by the plate is 70.22 V, with an error of 17.39%. The error is 17.62% when the wire is offset by 10 mm in the x and y directions. Under the same conditions, the number of plates is changed to six, the distance between each plate and the conductor is 50 mm, the average value of the initial floating potential sensed by the plate is 67.77 V, and the average value of the initial floating potential sensed by the plate is 67.51 V when the conductor is shifted by 10 mm in the x and y directions. The error is 0.38%. Through simulation analysis, it can be seen that the structure of the six-electrode array can effectively reduce the error caused by the change in wire position compared with the sensor with one electrode plate.

Figure 2 shows a schematic diagram of the structure of a single electrode of the sensor. Layers 1, 3, 5, and 7 represent electrode sensing layers with a thickness of 0.035 mm. Layers 2 and 4 represent FR4 materials with a thickness of 0.2 mm. Layer 6 represents FR4 with a thickness of 1.2 mm. The sensor unit is composed of two capacitors and such a structure is conducive to reducing external interference and improving sensor performance. The third and fourth layers are ground layers, which can shield external signal interference. In addition, the ground layer needs to be connected to the reference ground of the signal under test to provide an accurate reference potential.

According to the electric field coupled contactless voltage theory and simulation results, the overall structure of the sensor is designed. The non-contact voltage sensor with a multi electrode array designed in this paper is shown in Figure 3. The sensor includes two groups of capacitive voltage sensors installed on an electrically insulating support member. The first group of sensors is positioned along the inner closed path and the second group of sensors is positioned along the outer closed path, surrounding the inner closed path and connected in parallel. The wire to be measured can be placed on the central sensor of the sensor, so that the sensor surrounds the axis of the wire to be measured, each sensor has a signal electrode connected to the corresponding signal conductor, which can measure the voltage in the measured conductor and the voltage waveform on the reference conductor.

The size of the sensor coupling capacitance $C_{m}$ is related to the sensing distance of the line under test and the area of the line under test. Among them, the sensing distance is determined by the manufacturing process of the PCB board. In order to shield the vertical interference of the induction head and attenuate the interference of the horizontal direction, the area of the top ground layer in Figure 2 should be larger than that of the sensor head. Figure 4 shows the relationship between the shielding degree and the area of the shielding layer. This paper defines the interference degree as the percentage of the interference capacitance $C_{j}$ to the coupling capacitance $C_{j}$, where $C_{j}$ is the coupling capacitance between the side interference line and the sensor head. When the interference is strengthened on the side of the electric field coupling sensor, the interference degree of the surrounding lines to the sensor decreases with the increase in the area of the shielding layer. In order to ensure the good shielding effect and small size of the sensor at the same time, this paper selects Δx=3 mm. 

## 3. Research on Experiment and Data Processing Algorithm

### 3.1. Experimental Platform Construction and Data Collection

The schematic diagram of the non-contact voltage measurement system based on the multi-capacitor array is shown in Figure 5. A 50 Hz power frequency voltage is generated by a voltage source and the experimental comparison test is carried out using an Agilent 16-digit digital multimeter. The pico 5443D 16-bit PC oscilloscope is utilized to simultaneously record the waveforms of the voltage sensor and digital multimeter. This sensor mainly measures the rated voltage of 220 V, so it is measured within the range of 80–120% of the rated voltage specified by the IEC60044-7 standard. The power frequency experimental platform is shown in Figure 6.

In order to obtain the linearity of the voltage sensor, the output voltage of the DC-5 kV programmable AC source is adjusted while the distance between the sensor and the transmission line remains unchanged and the output voltage of the digital multimeter and the voltage sensor is recorded at the same time. The fitting curve of the experimental results is obtained, but before this, the experimental data need to be processed to obtain the most accurate measurement data.

### 3.2. Signal Processing Algorithms

To meet the performance requirements of a multi-sensor voltage measurement system, signal processing algorithms are key. The signal processing algorithm must not only ensure the accuracy of the measurement, but also be easy to implement, flexible, and adaptable. In a multi-sensor voltage measurement system, the measurement results mainly depend on the processing of the measurement signals of all sensors. The output of the voltage sensor represents the distribution of the space electric field around the wire, including the interference field signal introduced by other space electric fields and the noise caused by the uncertainty of the position of the voltage sensor itself. At present, the commonly used signal processing algorithms mainly include the mean value method, the spatial Fourier transform method, and the Kalman filter method.

The average value algorithm approximates the voltage value to be measured by summing the discrete measurement values, so the approximation error is closely related to the number of measurement values, that is, the number of sensors in the sensor ring. It can be seen that, only when the number of sensors is sufficient, the influence of the approximation error on the measurement results is small enough. Further, when the number of sensors is small, the measurement error will be very large. For the spatial Fourier transform, as the measurement error decreases exponentially with the number of sensors, high accuracy can be obtained without a large number of sensors. Even when the relative position of the interference voltage and the sensor ring is special, the influence of the interference voltage on the measurement accuracy can be completely eliminated in theory. Therefore, when the number of sensors is limited, the accuracy of the spatial Fourier transform is far better than the average algorithm. Kalman filter is a method to obtain the best filter value through signal estimation and correction based on the known system model and noise characteristics. The algorithm based on the Kalman filter can be regarded as an improvement of the average algorithm. As the object of averaging is the optimally filtered sensor output sequence, it can greatly reduce the measurement error of the averaging algorithm when the number of sensors is limited.

In summary, this paper uses the Kalman filter algorithm to process the signal. Kalman filter is a method used to obtain the best filter value by estimating and correcting the signal for the known system model and noise characteristics. This paper proposes a signal processing algorithm for a multi-sensor AC voltage measurement system based on Kalman filtering. According to the electric field distribution around the wire, the state equation of the system is established and the recursive method of Kalman steady-state filtering is provided. The algorithm based on Kalman filter can be regarded as an improvement to the traditional average algorithm. As the object of averaging is the optimally filtered sensor output sequence, the measurement error of the averaging algorithm when the number of sensors is limited can be greatly reduced. According to the analysis of the characteristics of the interference voltage, determining the appropriate filter parameters is the key to the success or failure of the Kalman filter algorithm. When the characteristics of the interference voltage are unknown, it is a necessary part of the measurement system to correct the Kalman filter by means of parameter estimation.

In a variety of applications, the structure of multiple parallel conductors is most common. In the modeling of multi-sensor AC voltage measurement, the double parallel straight wire model is often used. By studying the electric field distribution characteristics of double conductors, it is not difficult to generalize to more parallel straight conductors. As shown in Figure 7, there are two parallel long straight wires, where the distance is D, the magnitude of the voltage on the two wires is U, and the direction of the current is opposite. The distances between point P and the two wires are $r_{1}$ and $r_{2}$, respectively. When the distance between the wires and the radius of the wires is much smaller than the length of the wires, it can be approximated as the parallel plane electric field generated by two infinitely long straight wires.

Assuming that the linear charge density of a uniformly charged infinitely long straight wire is τ, the dielectric constant in a vacuum is $\varepsilon_{0}$, the point $P_{0}$ is selected as the reference point, the vertical distance between the point $P_{0}$ and the wire is $r_{0}$, and $\varphi \left(r_{0}\right)=0$. In Figure 7, the potential difference of two infinitely long straight wires at point P is as follows:(5)$$\varphi \left(r_{1}\right)=-\frac{\tau }{2\pi \varepsilon_{0}}\ln\frac{r_{1}}{r_{0}}$$
(6)$$\varphi \left(r_{2}\right)=\frac{\tau }{2\pi \varepsilon_{0}}\ln\frac{r_{2}}{r_{0}}$$

According to Formula (6) and Formula (7), the potential difference at point P can be obtained as follows:(7)$$\varphi \left(P\right)=\varphi \left(r_{1}\right)+\varphi \left(r_{2}\right)=\frac{\tau }{2\pi \varepsilon_{0}}\left(\ln\frac{r_{2}}{r_{0}}-\ln\frac{r_{1}}{r_{0}}\right)$$

Then,
(8)$$\varphi \left(P\right)=\frac{\tau }{2\pi \varepsilon_{0}}\ln\frac{r_{2}}{r_{1}}$$

The multi-sensor system in this paper adopts the sensor ring structure. For the convenience of calculation, Formula (9) is transformed into the expression in the cylindrical coordinate system:(9)$$\varphi \left(P\right)=\frac{\tau }{2\pi \varepsilon_{0}}\ln\frac{\sqrt{{\left(D-r\cos\varphi \right)}^{2}+{\left(r\sin\varphi \right)}^{2}}}{r}$$

Take the negative gradient of $\varphi_{P}$ as the following:(10)$$E_{P}=-\frac{\partial \varphi }{\partial r}=\frac{\tau }{2\pi \varepsilon_{0}}\left(1+\frac{3Dr\cos\varphi -2D^{2}-3r^{2}}{D^{2}+r^{2}-2Dr\cos\varphi }\right)$$

Assuming *D = βr*, then:(11)$$E_{P}=\frac{\tau }{2\pi \varepsilon_{0}}\left(1+\frac{3\beta \cos\varphi -2\beta^{2}-1}{\beta^{2}-2\beta \cos\varphi +1}\right)$$

Among them, $\left(3\beta \cos\varphi -2\beta^{2}-1\right)/(\beta^{2}-2\beta \cos\varphi +1)$ represents the influence of the interference electric field generated by the parallel wires. When the interference voltage value is not equal to the voltage value to be measured, and is K times the voltage value to be measured, the influence of the interference electric field is $K\left(3\beta \cos\varphi -2\beta^{2}-1\right)/(\beta^{2}-2\beta \cos\varphi +1)$.

The sensor structure designed in this paper is a ring structure and a voltage measurement system realized by multiple sensors is used, which can greatly reduce the difficulty of analysis and calculation. The sensor arrangement of the multi-sensor system is shown in Figure 8. A plurality of voltage sensors is evenly distributed on a ring of radius r.

According to the electric field distribution of the transmission wire, the output sequence $U_{\left(n\right)}$ of the voltage sensor can be obtained as shown in Formula (12). Taking the voltage $U_{1}$ to be measured as the system state quantity and the voltage sensor output sequence $U_{\left(n\right)}$ as the measurement value, the state equation and the measurement equation are established respectively. In the state equation shown in Formula (13), as the value of the voltage to be measured remains relatively unchanged, the system state remains unchanged. In the measurement equation shown in Formula (14), the measurement noise $z\left(n\right)$ is the influence of the interference electric field on the output of the voltage sensor.
(12)$$U_{\left(n\right)}=\frac{\tau }{2\pi \varepsilon_{0}}\left(V_{1}\left(n\right)+V_{2}\left(n\right)\frac{3\beta \cos\left(2\pi \frac{n}{N}\right)-2\beta^{2}-1}{\beta^{2}-2\beta \cos\left(2\pi \frac{n}{N}\right)+1}\right)$$
(13)$$V_{1}\left(n+1\right)=V_{1}\left(n\right)$$
(14)$$V_{\left(n\right)}=\frac{\lambda }{2\pi \varepsilon_{0}}V_{1}\left(n\right)+z\left(n\right)$$

Then,
(15)$$z\left(n\right)=\frac{\lambda }{2\pi \varepsilon_{0}}\cdot V_{2}\left(n\right)\frac{3\beta \cos\left(2\pi \frac{n}{N}\right)-2\beta^{2}-1}{\beta^{2}-2\beta \cos\left(2\pi \frac{n}{N}\right)+1}$$

The corresponding Kalman steady-state filter equation is as follows:(16)$$\left\{\begin{array}{c} {\hat{V}}_{1}\left(n\left|n\right.\right)={\hat{V}}_{1}\left(n\left|n\right.-1\right)+M\left(U\left(n\right)-\frac{\lambda }{2\pi \varepsilon_{0}}\right)\cdot {\hat{V}}_{1}\left(n\left|n\right.-1\right)\\ {\hat{V}}_{1}\left(n\left|n\right.+1\right)={\hat{V}}_{1}\left(n\left|n\right.\right) \end{array}\right.$$

Among them, ${\hat{V}}_{1}\left(n\left|n\right.-1\right)$ is the estimated value of the state quantity $V_{1}$ based on the first n−1 measurement values; ${\hat{V}}_{1}\left(n\left|n\right.\right)$ is the state variable based on the nth measurement value. For the estimated value of the quantitative update, M is the steady-state gain that minimizes the covariance of the estimated error for the system noise $Z\left(n\right)$. The properties of the system noise $z\left(n\right)$ determine the value of M.

By synthesizing the two Kalman steady-state filter equations, we can obtain the following:(17)$${\hat{V}}_{1}\left(n+1\left|n\right.\right)=\left(1-M\frac{\lambda }{2\pi \varepsilon_{0}}\right){\hat{V}}_{1}\left(n\left|n\right.-1\right)+M\hat{U}\left(n\right)$$
(18)$$\hat{U}\left(n\left|n\right.\right)=\frac{\lambda }{2\pi \varepsilon_{0}}\left(1-M\frac{\lambda }{2\pi \varepsilon_{0}}\right){\hat{V}}_{1}\left(n\left|n\right.-1\right)+M\frac{\lambda }{2\pi \varepsilon_{0}}\hat{U}\left(n\right)$$
where $\hat{U}\left(n\left|n\right.\right)$ is the optimal filtering result of the measured value. The filtered sensor output sequence is averaged and the result is used as the voltage value to be measured.

Linear fitting is performed on the processed experimental data, as shown in Figure 9; the expression of the fitting curve is y=3.0640x−2.0298 (x is the voltage of the transmission line, kV; y is the output voltage of the voltage sensor, mV). The results show that the voltage sensor has good linearity.

According to the measured voltage and actual voltage in Figure 9, it can be obtained that the theoretical voltage divider ratio $K_{n}$ of the voltage sensor is about 325.47 and the voltage reduction value $U_{c}$ can be expressed by the following formula:(19)$$U_{c}=K_{n}U_{d}$$

In order to evaluate the accuracy of the sensor more intuitively, it is necessary to calculate the relative error (δ%) [31] and phase error $\varphi_{u}$ of the sensor:(20)$$\delta \%=\frac{U_{c}-U_{l}}{U_{l}}\times 100\%$$
(21)$$\varphi_{u}=\varphi_{d}-\varphi_{t}$$
where $U_{l}$ is the voltage of the wire, $\varphi_{d}$ is the phase measured by the voltage probe, and $\varphi_{t}$ is the output phase of the Tektronix high-voltage probe. It can be seen from Table 1 that, within the 80–120% rated voltage range specified by the standard, the measurement error of the sensor is controlled within ±1% and the phase error is controlled within 4°. $U_{n}$ in Table 1 is the rated voltage and the rated voltage measured in this paper is 220 V.

The instantaneous value waveforms of the input and sensor output voltages are shown in Figure 10. It can be seen that their amplitudes and phase angles are basically the same and the sensor can highly recover the voltage to be measured.

## 4. Conclusions

Aiming at addressing the problems of the existing voltage sensors such as large volume, difficult installation, complex manufacturing process, high cost, and the fact that the relative position of the wire and the sensor has a great influence on the measurement accuracy, this paper proposes a multi-electrode array non-contact voltage sensor manufactured using PCB technology for the structure and a signal processing algorithm for a multi-sensor voltage measurement system. The experimental measurement of the developed sensor is carried out in the laboratory. The experimental and simulation results show that the ratio error is less than ±1% and the phase difference is less than 4° under the condition of power frequency 50 Hz. When the relative position of the wire and the sensor is in the direction of 45° and the offset is 10 mm, the relative error of the voltage sensor with single-electrode structure is 17.62%, while the relative error of the multi-electrode voltage sensor designed in this paper is only 0.38%. Compared with the single-electrode structure, the sensor structure proposed in this paper is effective. Errors caused by changes in the relative positions of wires and sensors are reduced. The sensor meets the development needs of smart grid measurement sensor intelligence, miniaturization, and convenience. The sensor can be used in low-voltage distribution networks; for example, for the measurement of 220 V single-phase line voltage. However, the sensor designed in this paper only measures the low voltage of the power frequency of 50 Hz. In the future, in-depth research can be carried out to reduce the error of the sensor, as well as increase its bandwidth and range, among others.

## Figures and Tables

**Figure 1 sensors-22-08573-f001:**
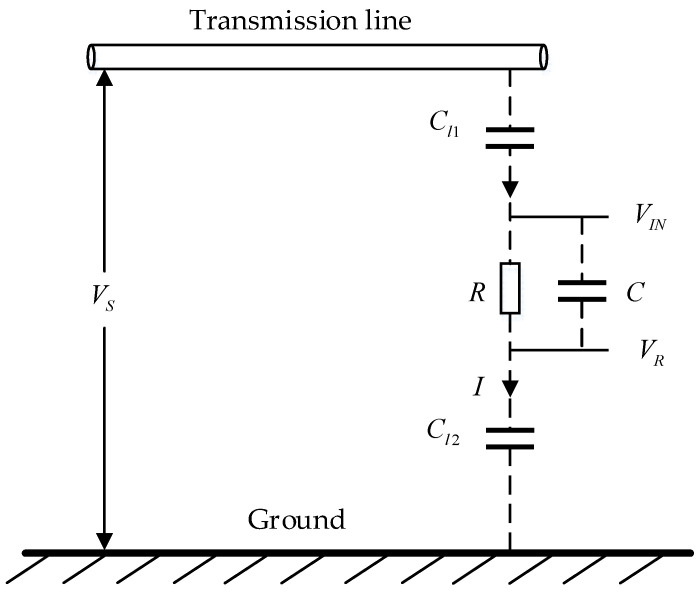
Measurement principle.

**Figure 2 sensors-22-08573-f002:**
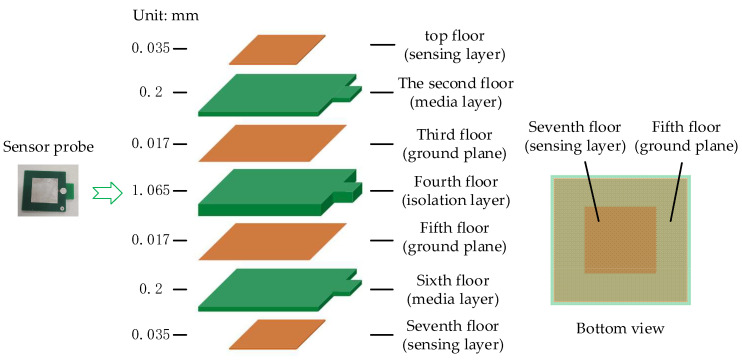
Structural diagram of the sensing electrode unit.

**Figure 3 sensors-22-08573-f003:**
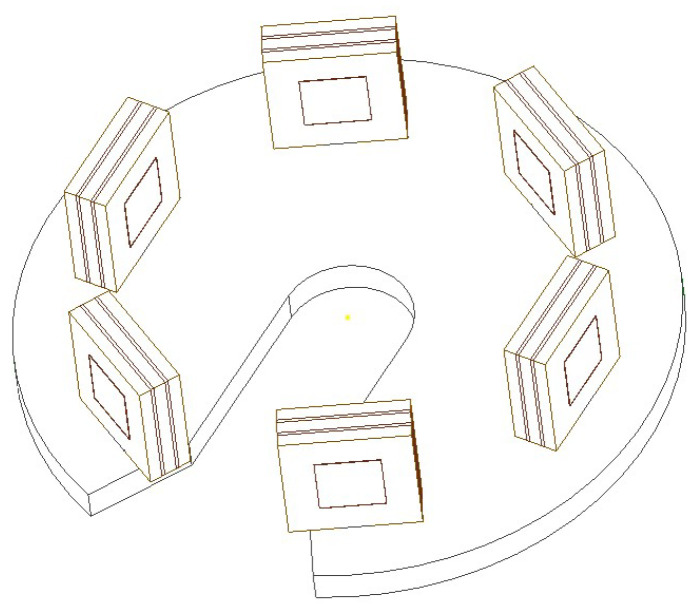
Schematic diagram of the three-dimensional structure of the sensor.

**Figure 4 sensors-22-08573-f004:**
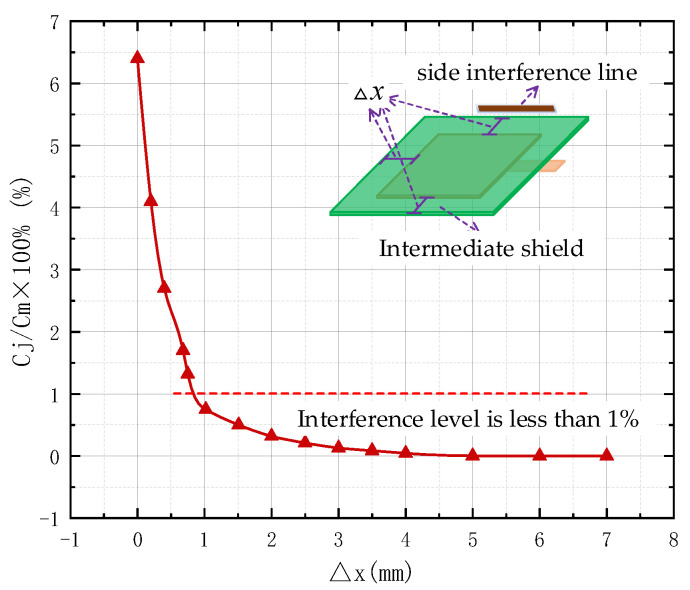
The relationship between anti-interference degree and shielding layer area.

**Figure 5 sensors-22-08573-f005:**
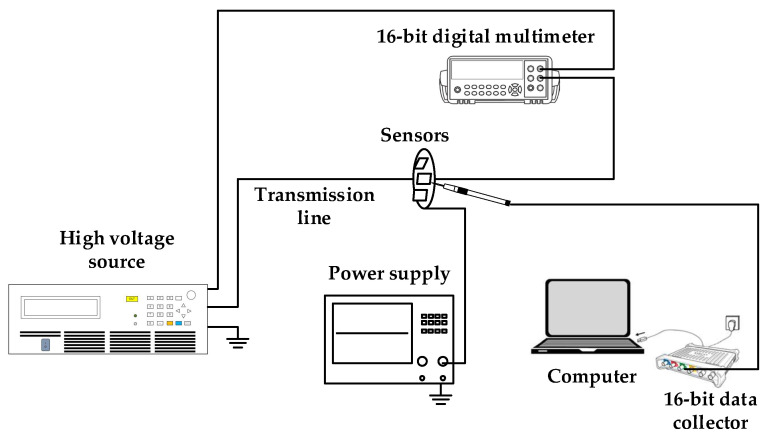
Schematic diagram of the measurement system.

**Figure 6 sensors-22-08573-f006:**
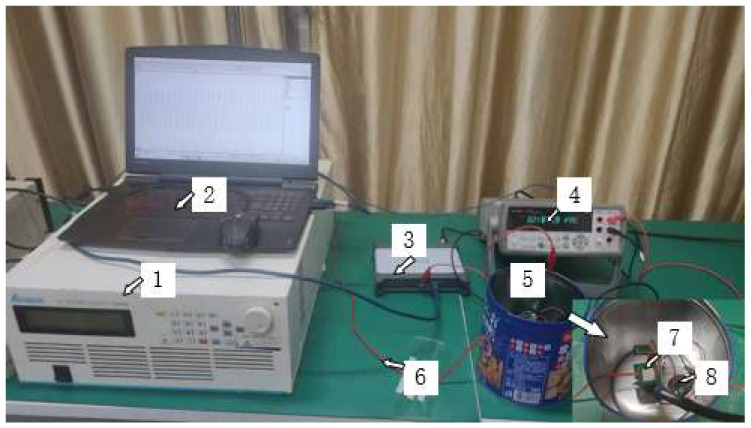
Power frequency experimental platform. (**1**)—high voltage source, (**2**)—PC terminal data acquisition, (**3**)—pico 5443D 16-bit PC oscilloscope, (**4**)—Agilent 16-bit digital multimeter, (**5**)—shield cover, (**6**)—wire to be tested, (**7**)—sensor, and (**8**)—9 V dry battery.

**Figure 7 sensors-22-08573-f007:**
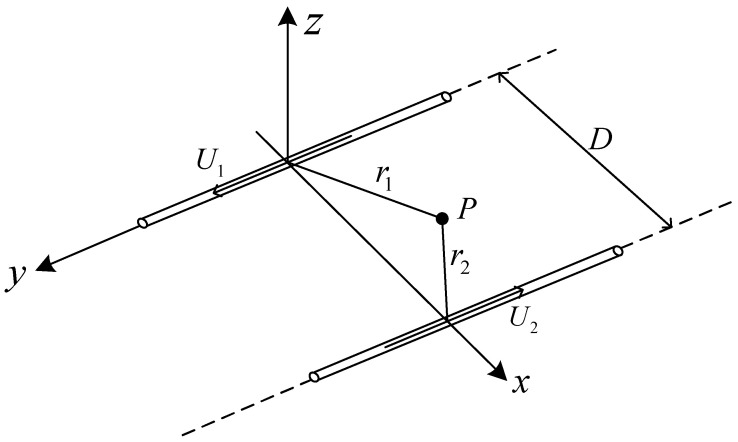
Schematic diagram of double parallel AC conductors.

**Figure 8 sensors-22-08573-f008:**
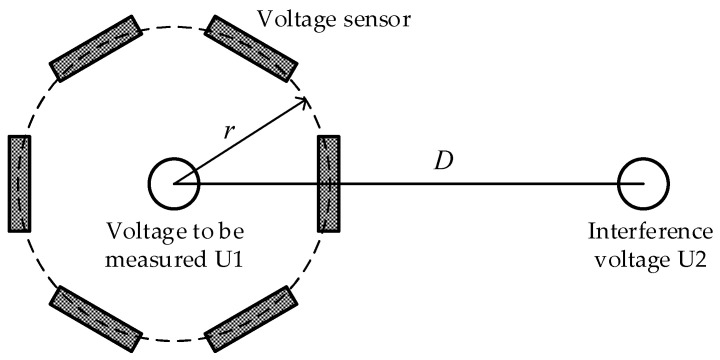
Schematic diagram of the sensor arrangement of the multi-sensor system.

**Figure 9 sensors-22-08573-f009:**
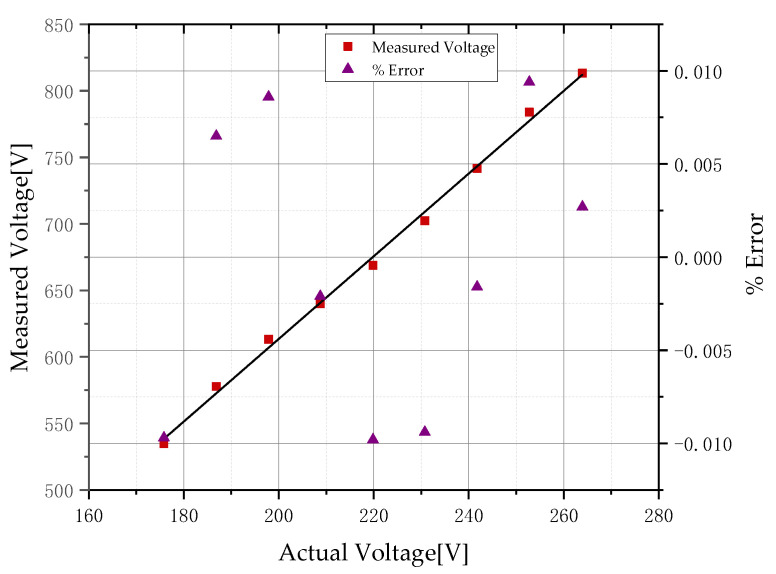
The measured voltage using the prototype and the ratio error characteristics.

**Figure 10 sensors-22-08573-f010:**
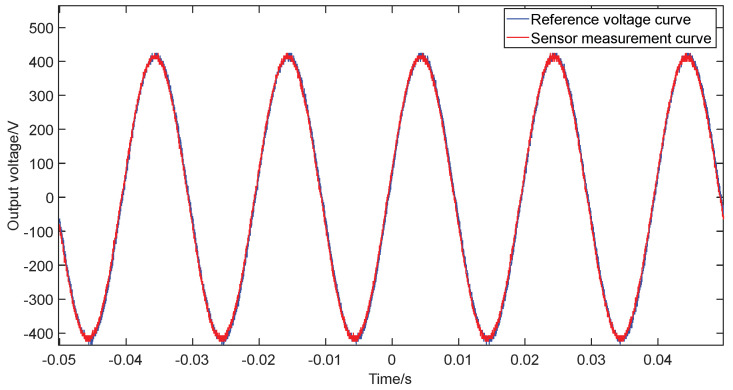
Momentary values of input (**red**) and sensor output (**blue**) voltage.

**Table 1 sensors-22-08573-t001:** Calculation results of the sensor measurement value ratio difference and angular difference.

Measuring Point	$U_{l}/V$	$U_{s}/\mathrm{mV}$	ε/%	$\varphi_{u}/\left({}^{\circ}\right)$
80%$U_{n}$	175.82	535	−0.97%	3.12
85%$U_{n}$	186.85	578	0.65%	2.79
90%$U_{n}$	197.87	613	0.86%	3.41
95%$U_{n}$	208.76	640	−0.21%	3.16
100%$U_{n}$	219.84	669	−0.98%	2.84
105%$U_{n}$	230.76	702	−0.94%	2.61
110%$U_{n}$	241.78	742	−0.16%	2.53
115%$U_{n}$	252.77	782	−0.94%	3.18
120%$U_{n}$	263.95	813	0.27%	3.12

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
