# Peer review of "Research on a Non-Contact Multi-Electrode Voltage Sensor and Signal Processing Algorithm"

_sensors, 2022, doi:10.3390/s22218573_

Round 1

Reviewer 1 Report

Dear Authors,

I have some comments on your article:

1. Literature should be checked if there are no newer items. Especially from the last 18 months.

2. At the end of the introduction section, there is no information on how the article is organized.

3. All indexes in symbols in text and equations should be checked carefully.

4. To what extent could the improvement of the PCB production process still improve the accuracy of the measurement?

5. It would be good to perform a series of tests on sensors performed, for example, by various contractors.

6. The summary should contain a few words about the possibility of using this type of smart sensor in useful measurements.

Reviewer 2 Report

The authors proposed a novel multi-sensor voltage system. The paper was well-divided and well-referenced. Therefore, those are my queries and comments.

  • The abstract is confusing. The text is excessively straightforward and lacks information. I suggest the authors review the abstract section and improve the text. For example, describe better the research gap you have filled.

  • Highlight the comparison with traditional techniques. Use statical metrics to compare your results with other methods, especially in the result section and Conclusion.

  • The quality and resolution of Figures 9 and 13 must be improved.

  • The phrases used in the Conclusion section are very similar to other parts of the text. Therefore it is very repetitive. The authors should review the text of the conclusion.

Round 2

Reviewer 1 Report

Dear Authors,

Thank you very much for introducing changes that have improved the quality of the article. I have no more comments.

Best regards